# Applying Principles from Prevention and Implementation Sciences to Optimize the Dissemination of Family Feeding Interventions

**DOI:** 10.3390/ijerph17197038

**Published:** 2020-09-25

**Authors:** Louise Parker, Alexandra Jackson, Jane Lanigan

**Affiliations:** 1Extension Family Development, Human Development, Washington State University, Seattle, WA 98103, USA; 2Human Development, Washington State University, Vancouver, WA 98686, USA; Alexandra.m.jackson@wsu.edu (A.J.); jlanigan@wsu.edu (J.L.)

**Keywords:** implementation, dissemination, family feeding interventions, prevention science, nutrition

## Abstract

Because families are the primary food socialization agent for children, they are a key target for nutrition interventions promoting healthy eating development. Although researchers and clinicians have developed and tested successful family nutrition interventions, few have gained widespread dissemination. Prevention and implementation science disciplines can inform the design, testing, and dissemination of feeding interventions to advance the goals of widespread adoption and population health impact. We review concepts and frameworks from prevention science and dissemination and implementation (D&I) research that are useful to consider in designing, implementing, and evaluating feeding interventions. Risk and protective factor frameworks, types of translation processes, and implementation dimensions are explained. Specifically, we address how research–practice partnerships can reduce time to dissemination, how designing for modularity can allow for contextual adaptation, how articulating core components can strengthen fidelity and guide adaptation, and how establishing technical assistance infrastructure supports these processes. Finally, we review strategies for building capacity in D&I research and practice for nutrition professionals. In sum, the research and knowledge bases from prevention and implementation sciences offer guidance on designing and delivering family interventions in ways that maximize the potential for their broad dissemination, reducing time to translation and optimizing interventions for real-world settings.

## 1. Introduction

Childhood is recognized as a critical period for establishing healthy eating behaviors. Nutrition knowledge, preferences, and behaviors formed during this time often persist into later childhood, adolescence, and adulthood [1,2]. Children begin constructing the conceptual system that will guide their eating behaviors primarily through direct experience with food/feeding [3], and family is the primary food socialization agent [4]. Healthy eating in childhood supports growth and development and reduces the risk of negative health outcomes [5,6].

Thus, the family is a key target for public health nutrition interventions that promote healthy child eating development. Although breastfeeding promotion has been successfully implemented in many parts of the world [6], interventions that target child eating development after age two are less prevalent. Among the family nutrition interventions developed by researchers and clinicians that have undergone rigorous efficacy and/or effectiveness trials, few have gained widespread dissemination. This article describes how the prevention and implementation science disciplines can inform the design, testing, and dissemination of family-focused feeding interventions to advance the goals of widespread adoption and population health impact. As interdisciplinary fields that focus on closing the research-to-practice gap, frameworks from prevention and implementation science offer guidance for optimizing the impact of interventions in real-world settings.

In this narrative review, we highlight seminal research describing prevention and implementation science and suggest four practices to maximize dissemination and implementation: creating research–practice partnerships, designing for modularity, articulating core components, and establishing technical assistance to increase dissemination and implementation of interventions. We also illustrate each practice with examples from a recently developed family feeding intervention. In the final section, we propose structural changes to engage researchers and train front-line providers in dissemination and implementation strategies to optimize family feeding interventions for community-level benefit.

## 2. Prevention Science

Prevention science includes the etiology of a problem, determining efficacious and effective interventions (often referred to as programs or policies) to prevent the problem, and the scaling-up or dissemination of tested interventions to support improved public health. To assess and describe the etiology of the problem, prevention science utilizes a risk and protective factor framework. Risk factors precede and are associated with a negative outcome, and can occur at multiple levels (e.g., individual, family, community, cultural) [7]. Risk factors are often correlated with one another and have a cumulative effect on the development of negative outcomes. Alternatively, protective factors are associated with a lower likelihood of the outcome [7]. Risk and protective factors can be fixed or variable, affect multiple outcomes, and may influence one another over time [8]. The risk and protective factor framework is then used to inform interventions that target variable risk and/or protective factors to promote health and reduce negative outcomes. For example, controlling (or restrictive) food parenting practices is a malleable risk factor, while the parental encouragement of a variety of foods, meal and snack routines, and attention to hunger and fullness cues are protective factors that may be incorporated in family feeding interventions [9,10].

The inclusion of risk and protective factors in preventive interventions includes a process referred to as Type 1 and Type 2 translation [11]. Type 1 research applies the risk and protective factor framework in the development of interventions, through tightly controlled, rigorous efficacy trials (i.e., could the program work?). Type 2 translation focuses on the processes and mechanisms in which interventions are integrated on a large scale, including effectiveness trials (i.e., does the program work?), dissemination and implementation (D&I) (i.e., making a program work, also referred to as scaling-up) [11,12]. Within each step of testing evidence-based programs, Gottfredson et al. [13] has developed a set of standards of evidence for efficacy, effectiveness, and scaling-up of evidence-based interventions. Research on Type 2 translation, or the dissemination of programs to support community benefits, is limited and a relatively new area of exploration.

## 3. Implementation Science

A central concern of implementation science, also referred to as dissemination and implementation (D&I) research, is shortening the time it takes to translate research to practice. Evidence suggests that it can take more than 17 years to see widespread benefits from health-related interventions [14], but strengthening strategies for the D&I of effective programs could decrease this gap. Dissemination includes the distribution of a targeted intervention to a specific audience, whereas implementation involves the use of strategies to adopt and integrate evidence-based interventions in specific settings [12]. The goal of D&I is to improve the adoption, adaptation, delivery, and sustainability of effective interventions across differing intervention settings [12]. To meet this goal, implementation science assesses the implementation of the intervention, including concepts such as what aspects of the intervention are delivered and how well the program was conducted, which in turn provides additional information about the theory behind the intervention [15].

Eight dimensions of implementation have been articulated, including fidelity, dosage, quality, participant responsiveness, program differentiation, program reach, adaptation, and program monitoring [15,16]. This review will focus on three dimensions: dosage, fidelity, and adaptation. Dosage describes the amount of the program that is delivered [16]. Fidelity indicates the extent to which the intervention provided corresponds with the original intervention. Adaptation refers to changes made to the intervention during implementation [16]. Fidelity focuses on rigid adherence to the intervention, thus minimizing the need or the opportunity for adaptation [17,18].

The assessment of the dosage and/or fidelity of an intervention during implementation is critical to determine if the program is delivered as intended and is associated with positive outcomes. While the assessment of program implementation is necessary to indicate relationships between the intervention and outcomes, the measurement of implementation is relatively new, with most studies identifying a single dimension of implementation [16]. Data from over 500 studies that included at least one measurement of implementation indicates that few evidence-based interventions are implemented at 80% of their intended dosage or fidelity, and most interventions are implemented at 60% of what was intended [15]. While measures of implementation indicate low levels of adherence to the intervention as designed, the assessment of dimensions of implementation is associated with better outcomes [15]. A recent meta-analysis of obesity interventions indicated that evidence-based interventions are only 75% as effective when scaled up (disseminated) compared to the benefits reported in efficacy trials [19]. These findings highlight the importance of shifting the focus from internal validity (that occurs in efficacy and effectiveness trials), to external validity in D&I [12].

Fidelity and adaptation are often viewed as opposing ends of a continuum, with adaptation leading to a loss in fidelity and thus failure to replicate outcomes. However, based on the data from Durlak and DuPre [15] indicating that adaptation may be inevitable in preventive interventions (most interventions reporting implementation of the intervention at 60% of what was intended), pre-planned adaptation could reduce the tension between fidelity and adaptation while supporting intervention effectiveness [18]. For example, Kemp [18] used the analogy of a basic plain cake recipe with variations (chocolate, coconut, apple, marble, etc.). The analogy describes the use of core components, or the essential kernels of the intervention, and planned adaptation to balance fidelity and adaptation in the Maternal Early Childhood Sustained Home-visiting (MESCH) program. The plain cake recipe (core components) is established through rigorous testing and includes the necessary ingredients, methods, and equipment, with cake variations (adaptations) that are determined in advance. Decisions to include variations are made as part of a reflexive partnership with the intervention developers and technical advisors, with the aim of enhancing the program (rather than substitution during program implementation).

## 4. Application of Principles from Implementation/Dissemination

Addressing and reducing barriers across the process of translation may decrease the length of time for evidence-based interventions to achieve community-level benefits and optimize investments in the development of family feeding interventions. These efforts include the consideration of what works for whom and under what conditions, the identification of relevant interventions, and subsequent matching to ensure the best fit. Advances in the assessment of eating and dietary intake such as ecological momentary assessment provide the developers of interventions with a more nuanced understanding of contextual factors that influence intervention effectiveness and improve fit [20]. Developing a portfolio of programs, identifying core components, and allowing for adaptation and fidelity will support the next steps in the D&I of interventions to have wide-scale population impact.

The research and knowledge bases from prevention and implementation sciences can offer guidance on designing and delivering family interventions in ways that maximize the potential for their broad dissemination. Specifically, we address how research–practice partnerships can reduce the time to dissemination, how designing for modularity can allow for contextual adaptation, how articulating core components can support fidelity and guide adaptation, and how establishing technical assistance infrastructure can support these processes. We illustrate the application of these concepts in a recently developed and tested nutrition intervention—Food Feeding and Your Family (FFYF) [21].

Project teams that incorporate **research–practice partnerships** are more likely to create interventions that have higher rates of program adoption, implementation, and maintenance, compared to those utilizing a typical efficacy-effectiveness pipeline model [22]. When researchers and practitioners work together through every phase of intervention development, testing, and delivery, the resulting programs may be more relevant and practical for real-world delivery [23]. The FFYF team included childhood obesity and parenting researchers, nutrition curriculum specialists, and nutrition education supervisors and educators. The team was formed at the point of crafting the funding proposal and continued to collaborate through the stages of curriculum design, selection of research measures and data collection strategies, delivering the intervention, and collecting post-program qualitative data from facilitators and participants on the effectiveness of content and delivery. The team met monthly to facilitate ongoing communication between researchers and staff supervising program delivery, ensuring that implementation and data collection strategies could be refined regularly as unexpected challenges arose.

**Modularity** is another principle in intervention design that facilitates more widespread implementation and dissemination. It refers to developing interventions such that particular components can be easily separated and recombined [24]. This design feature allows organizations and service structures to incorporate and adapt specific components that are appropriate for their context and population. While more common in clinical interventions [25], prevention researchers are now focusing on modularity as another promising mechanism to promote the targeted scale-up of interventions. In the case of the FFYF intervention, members of the research team had previously developed Strategies for Effective Eating Development (SEEDs) with a modular design that included parent, child, and family components [26]. The FFYF team was able to adapt a series of SEEDs videos originally developed for parents of preschoolers to teach the same content to an audience of parents with children ages 2 to 8. The FFYF team also designed their intervention with modularity by developing and testing both face-to-face and online delivery versions. Parents in the traditional delivery version viewed the videos and participated in related small group activities, while online participants viewed a weekly video segment and completed an interactive web-based gaming activity. In preliminary analyses of program outcomes, researchers found few differences in effectiveness between the two options [27]. The addition of an online delivery option and gaming activities to FFYF aligned with research indicating that over 90% of parents were interested in online delivery and the option to engage their children with content [28]. This greatly expanded the modularity of FFYF, allowing practitioners to choose the delivery option that suits their context and audience or pair the online FFYF with the SEEDs child intervention, maximizing the potential for broader dissemination of both interventions.

**Clarity of core components** is a principle that encourages the developers of interventions to be clear about both the content and processes that are foundational in order to achieve the program’s intended outcomes. The use of risk and protective factor frameworks can be a useful tool in clarifying the essential ingredients of an intervention. When core components are ambiguous, practitioners struggle to make adaptation and implementation decisions that will not compromise program effectiveness [29]. In the case of FFYF, the adaptation of the SEEDs parent videos insured fidelity to the core content components of the intervention, as they were embedded in the video sessions [26], and the efficacy of using only the parent component of the family-based SEEDs intervention was evaluated.

The specification of core process components is particularly relevant for family-based interventions, which typically include both parent and child activities and delivery systems. A common adaptation issue in the dissemination process is the temptation to decouple family-based delivery and offer only the parent or child versions of the intervention or change the implementation in another significant way. Using Kemp’s [18] analogy discussed previously, adaptation decisions are sometimes made that leave out key ingredients in the plain cake recipe in the process of adding variations for new settings or audiences. One approach to providing clarity around the core process component of family-based delivery is to test a youth- or parent-only version of the intervention in an effectiveness trial. For instance, a feeding intervention focused on media literacy included the testing of a youth-only version of its family program in the research design. Researchers found that cognitive and behavioral outcomes for participants in the youth-only version were not as robust as those for youth in the family delivery model [30]. These findings provide clear information on core process components to practitioners considering adaptation of the intervention to new settings. A final advantage of identifying core components is that “minimal effective interventions” can be developed. A team of researchers reviewing early childhood obesity prevention trials recommend paring down the content of interventions to include only what is most closely tied to successful outcomes in research trials. The resulting program is the most minimal intervention that can make a difference. The authors suggest that these streamlined versions of an intervention may be more easily implemented and disseminated [31].

**Technical assistance** needs should be considered during the intervention design and evaluation phases as well as the D&I phase and included in the timeline and budget. Inadequate technical assistance has been shown to reduce intervention effectiveness [32] and hinder upscaling [33]. Family-based nutrition interventions that provide modularity and use a core components approach to maximize stakeholder flexibility involve greater decision-making around adaptations. In order to assist in the D&I phase, intervention developers need to provide technical assistance to guide programs in making informed decisions about planning and delivering these interventions. The complexity of the family-based nutrition intervention, the resources and capacity of the intervention developer, and organizational capacity within implementing systems will influence technical assistance needs. The process evaluation and personnel training conducted during efficacy and effectiveness trials provide valuable information about training topics and delivery, frequently asked questions, implementation challenges, and effective practices. The FFYF intervention engaged in intensive training of Expanded Food and Nutrition Education Program (EFNEP) nutrition educators and extensive process evaluations during the effectiveness study [34] which are being used to identify likely technical assistance needs in the dissemination phase.

Technical assistance includes an intervention guide, training program staff, and ongoing support, which can include site visits. An intervention guide is a critical reference that needs to be available in all languages for which the intervention was developed. Training improves intervention delivery and outcomes [32,33], but can become a barrier to scaling up when intervention developers do not have the infrastructure to deliver or sustain in-person training, or when implementing programs do not have the resources to support in-person training. A recent study of the Out-of-school Nutrition and Physical Activity program implementation compared an online and an in-person train-the-trainer approach during the program D&I, and found that the in-person training was more effective at increasing achievement of program goals, but cost twice as much per person as online training. Lee et al. [35] concluded that online training may be a viable alternative. In contrast, Benjamin et al. [36] found in-person and web-based training to be equally effective. Technical assistance can also include ongoing support in the form of site visits or online help. Provision of this level of technical assistance requires ongoing resources for personnel during the D&I phase, which is not supported in traditional grant funding mechanisms. Models for providing technical support include partnerships with public or private organizations who have infrastructure for D&I (e.g., Eating Smart, Being Active [37]) and the formation of an organization to provide ongoing support for the intervention (e.g., Coordinated Approach to Child Health (CATCH) [38]).

## 5. Capacity Building for Dissemination and Implementation

To build capacity in D&I, it is essential to engage both researchers and practitioners in the development and use of effective interventions [39]. A mixed-methods study of international researchers in nutrition and physical activity identified barriers to D&I research including a lack of expertise, lack of organizational support or value for D&I, methodological challenges, funding priorities, academic performance structures, and an embedded culture that is not conducive to translational research [40]. To reduce these barriers and increase the real-world impact of effective interventions, Koorts et al. [40] suggest that funders prioritize supporting long-term research that is implementable and can be scaled up, that academic institutions restructure performance metrics to place higher value on translational research, that organizations increase employment opportunities that bridge academia, practice, and policy, and that training programs for practitioners and policymakers better integrate research and practice.

The engagement of practitioners is a key element in supporting the effective implementation and dissemination of evidence-based interventions. Front-line providers require training to understand the importance of D&I, as well as the skills and strategies to implement the interventions. In the United States, core competencies for nutrition educators from the Society for Nutrition Education and Behavior (SNEB) and the Accreditation Council for Education in Nutrition and Dietetics (ACEND) focus on program development and evaluation, including the use of theory to inform the development of programs to support dietary behavior change, process and outcome evaluations, and addressing community needs during program implementation [41,42,43]. To better prepare front-line providers, we suggest incorporating competencies that include training and education of strategies to support D&I in addition to program development and evaluation. Front-line providers need to be aware of existing interventions, and may benefit from working with research teams to utilize a portfolio of programs as well as incorporate strategies from D&I to optimize family feeding interventions.

Potential resources that could support front-line providers in program implementation and the development of D&I competencies include the International Society for Behavioral Nutrition and Physical Activity (ISBNPA) Implementation, Translation, Scale-Up and Sustainability Special Interest Group [44] and partnering with implementation science organizations including the Society for Prevention Research, the National Implementation Research Network [45], and the Society for Implementation Research Collaboration [46]. The Canadian-based Center for Implementation also offers a detailed framework of Core Competencies for Implementation Practice that could be used to guide the development of academic courses and professional development for those in program delivery roles [47].

## 6. Conclusions

The merging of prevention and implementation science with the nutrition field can support the D&I of family feeding interventions, reducing time to translation and optimizing interventions for real-world settings. We suggest the inclusion of a risk and protective factor framework, along with the terminology and evidence-based methods used in implementation science to disseminate family feeding interventions. Specifically, we suggest the use of research–practice partnerships, modularity, clarity of core components, and technical assistance as strategies to address implementation issues including dosage, fidelity, and adaptation. Lastly, to build capacity in D&I research, we suggest structural changes to encourage translational research and increased training of practitioners to integrate research and practice.

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
