# Peer review of "Applying Principles from Prevention and Implementation Sciences to Optimize the Dissemination of Family Feeding Interventions"

_ijerph, 2020, doi:10.3390/ijerph17197038_

Round 1
Reviewer 1 Report
The manuscript “Applying principles from prevention and implementation sciences to optimize dissemination of family feeding interventions, makes interesting suggestions.
However, it is not explained on the basis of which methodology these suggestions are developed. The work seems to be based on a literature review, but the criteria considered to perform the search and selection of the included literature are not indicated.
Furthermore, in several parts of the work the focus of the study (family feeding interventions) is lost, so the suggestions could be applicable to any type of intervention. Authors should focus on answering the objective of their study
It is also necessary for the authors to be more specific in their suggestions, for example on page 5 line 227 the authors indicate “to build capacity in D&I, it is essential to engage both researchers and practitioners…”. I think many of us agree that this suggestion is no longer a new need, but the authors should be able to say how to implement this and the rest of the suggestions they make.
In addition, the authors must clearly indicate in the work what their contribution to knowledge in the area is.
Author Response
Responses to Reviewer 1 Comments
The manuscript “Applying principles from prevention and implementation sciences to optimize dissemination of family feeding interventions, makes interesting suggestions.
However, it is not explained on the basis of which methodology these suggestions are developed. The work seems to be based on a literature review, but the criteria considered to perform the search and selection of the included literature are not indicated.
Response: Our manuscript was not intended as a systematic review, but rather a short communication (which we view as a commentary or narrative review). Because it is not a systematic review, we did not include a methods section describing criteria for inclusion, etc. We have added a paragraph to the end of the Introduction section (lines 48-55) to clarify the purpose and content of the manuscript for readers.
Furthermore, in several parts of the work the focus of the study (family feeding interventions) is lost, so the suggestions could be applicable to any type of intervention. Authors should focus on answering the objective of their study.
Response: Thank you for this suggestion that helped us more effectively focus on examples from feeding interventions. We clarified in the introduction that we would illustrate each practice in the application section with an example from a recently developed family feeding intervention (line 52). We also removed the substance abuse intervention example from the discussion of core components (lines 194-200) and instead added additional content to two feeding intervention examples (lines 201-213).
It is also necessary for the authors to be more specific in their suggestions, for example on page 5 line 227 the authors indicate “to build capacity in D&I, it is essential to engage both researchers and practitioners…”. I think many of us agree that this suggestion is no longer a new need, but the authors should be able to say how to implement this and the rest of the suggestions they make.
Response: Thank you for pointing out the lack of detail in this section. We have added content in the capacity building section that more clearly articulates roles of front line practitioners in D&I and suggests practices to strengthen their capacity (lines 257-259, 266-269).
In addition, the authors must clearly indicate in the work what their contribution to knowledge in the area is.
Response: We are hopeful that our addition of a last paragraph to the introduction clarifies the contribution of the manuscript (lines 48-55).
Reviewer 2 Report
The aim of the manuscript - to review key concepts from prevention science and dissemination and implementation research that are useful in feeding interventions – is very interesting and an important issue as highlighted by Authors. Thus, I think that the manuscript is relevant and of interest to the readership and especially for public health professionals. However, the paper needs additional work in order to make it more accessible and understandable to the reader.
I suggest to include a specific comment on the potential benefit of the usage of new technologies for the assessment of nutritional habits - as Ecological momentary assessment (EMA) and to guide preventive intervention.
Abstract: this section requires a brief discussion/conclusion to summarize main evidence from the review.
Introduction section: in this section the objective(s) of the study is not stringent and clear and should be better stated. Authors are invited to better define, as done in the abstract section, the aim of the manuscript.
I suggest to include a paragraph on methods used to conduct the present review.
Author Response
Reviewer 2 Responses
The aim of the manuscript - to review key concepts from prevention science and dissemination and implementation research that are useful in feeding interventions – is very interesting and an important issue as highlighted by Authors. Thus, I think that the manuscript is relevant and of interest to the readership and especially for public health professionals. However, the paper needs additional work in order to make it more accessible and understandable to the reader.
Response: Thank you for recognizing the relevance of our manuscript. We have made revisions in all sections to add examples and explanations to improve clarity for the reader.
I suggest to include a specific comment on the potential benefit of the usage of new technologies for the assessment of nutritional habits - as Ecological momentary assessment (EMA) and to guide preventive intervention.
Response: Thank you for this suggestion. We have added a reference to EMA and its potential relevance for improving intervention effectiveness and fit in the application section (lines 132-134). We also added more detail to our discussion of online delivery as a technological approach to broadening dissemination in the modularity discussion (lines 173-175).
Abstract: this section requires a brief discussion/conclusion to summarize main evidence from the review.
Response: We have added a concluding line to the abstract (line 24-27) and edited to meet the maximum word count.
Introduction section: in this section the objective(s) of the study is not stringent and clear and should be better stated. Authors are invited to better define, as done in the abstract section, the aim of the manuscript.
We have added a paragraph to the end of the Introduction section to clarify both the purpose and content of the manuscript (lines 48-55).
I suggest to include a paragraph on methods used to conduct the present review.
Response: As we now clarify in the Introduction, the manuscript is not intended as a systematic review so we did not include a methods section with inclusion criteria, etc. We are submitting the paper as a short communication which we view as a commentary or narrative review, the term we now use in the introduction to clarify our purpose to the reader (line 48).
Round 2
Reviewer 1 Report
The authors did a good job improving the manuscript.